# Seasonal Variations on Testicular Morphology, Boar Taint, and Meat Quality Traits in Traditional Outdoor Pig Farming

**DOI:** 10.3390/ani14010102

**Published:** 2023-12-27

**Authors:** Sofia Botelho-Fontela, Sílvia Ferreira, Gustavo Paixão, Ricardo Pereira-Pinto, Manuela Vaz-Velho, Maria dos Anjos Pires, Rita Payan-Carreira, Luís Patarata, José M. Lorenzo, José A. Silva, Alexandra Esteves

**Affiliations:** 1Animal and Veterinary Research Centre (CECAV), AL4AnimalS, University of Trás-os-Montes and Alto Douro, 5000-801 Vila Real, Portugal; silviaferreira@utad.pt (S.F.); gustavopaixao@esa.ipvc.pt (G.P.); apires@utad.pt (M.d.A.P.); lpatarat@utad.pt (L.P.); jasilva@utad.pt (J.A.S.); alexe@utad.pt (A.E.); 2CISAS—Center for Research and Development in Agrifood Systems and Sustainability, Instituto Politécnico de Viana do Castelo, 4900-347 Viana do Castelo, Portugal; rpinto@ipvc.pt (R.P.-P.); mvazvelho@estg.ipvc.pt (M.V.-V.); 3CHRC—Comprehensive Health Research Centre, Department of Veterinary Medicine, University of Évora, Pole at Mitra, 7002-554 Évora, Portugal; rtpayan@uevora.pt; 4Centro Tecnológico de la Carne de Galicia 4, 32900 San Cibrao das Viñas, Spain; jmlorenzo@ceteca.net

**Keywords:** boar taint, seasonal impact, testicular morphometry, meat quality traits, reproductive seasonality, Bísaro boars

## Abstract

**Simple Summary:**

This article focuses on the season effect on testicular morphometry and meat quality from outdoor-reared Bísaro pigs. Bísaro, an autochthonous Portuguese breed, is raised for long periods in outdoor pens after puberty and slaughtered around one year old for high-quality meat products. A total of 20 Bísaro boars, aged 13 months at the time of slaughter, were distributed into two groups according to the season at slaughter: Wi (winter) with 9 participants and Su (summer) with 11 participants. Surprisingly, while testicular volume in the Su group suggested lower functionality, boar taint compound analysis showed no significant differences. However, the panelists detected a more pronounced aroma and flavor of boar taint in the summer group. The meat quality traits presented no marked differences, although the fatty acid profile was higher in Wi than in Su. This study suggests the existence of reproductive seasonality in male Bísaro pigs, hinting at a dynamic variation of boar taint compounds throughout the year.

**Abstract:**

Traditional outdoor pig farming is renowned for its emphasis on animal welfare and the production of highly valued, quality meat. While seasonality is known to impact certain animals, particularly those raised outdoors, there is a lack of research on Bísaro boars, a native Portuguese breed. This research study was conducted on a total of 20 male entire Bísaro pigs, reared in outdoor pens from 4 to 13 months old, and subsequently slaughtered. The animals were divided into two groups: one slaughtered in winter (Wi, *n* = 9), and the other in summer (Su, *n* = 11). The objective was to evaluate testicular morphometry, boar taint compounds, and meat quality traits, including sensory analysis and fatty acid profile. Testicles from the Su group exhibited reduced volume, indicating diminished functionality during that season. While no significant differences were observed in the boar taint compound analysis, panelists could discern a more intense aroma and flavor of boar taint in the Su meat. Other meat quality traits showed no significant variations, but the fatty acid profile displayed higher values in the Wi group. This study reveals that Bísaro boars experience reproductive seasonality, leading to variations in boar taint compounds across the seasons. This information is crucial for farm planning.

## 1. Introduction

Traditional outdoor pig production offers numerous benefits to both animals and consumers [1,2,3,4]. This method of rearing is widely regarded as favorable for animal welfare, as it allows pigs to express their natural behaviors more easily [1]. In recent years, there has been increasing scrutiny of intensive production systems, with consumers showing a heightened preference for alternative methods, prioritizing environmental and animal-friendly practices [5]. Such alternative systems encompass various elements, including outdoor access, free-range environments, natural resource-based feeding, and the utilization of indigenous pig breeds [6,7]. Autochthonous breeds, adapted to local environmental conditions and possessing greater disease resistance [8], have gained popularity. This approach aligns with eco-friendly farming principles and also has a positive impact on meat quality [5,9,10]. Indigenous pig breeds, often described as “rustic”, are recognized for their slow growth and are typically raised to a more advanced age [11,12]. Another trait of alternative systems is that the diet also includes native vegetation and fruits. Acorns are the most important and thoroughly studied feed ingredient in breeds such as Bísaro, Alentejano, and Iberian pigs [5,10,13]. Acorns are used to fatten the pigs and to impart their distinctive chemical characteristics to the pig’s muscle and adipose tissues [6]. Because they enhance the fat content and the fatty acid profile, important differences exist in flavor and juiciness [5]. However, the availability of acorns is seasonal, making the production of pork with these characteristics impossible throughout the entire year [6].

The seasonal variation in feed source availability is only one aspect of the changes in pig meat quality, composition, and flavor. The time of the year also drives physiological seasonality. The link between wild boars and domestic pigs regarding seasonal variations in reproductive and endocrine functions is evident [14,15,16]. Seasonal influences on testicular steroids are common among wild boars. The seasonal influences on the gonadal function are related to differences in melatonin secretion, which determines LH secretion’s secretory pattern. While the domestic pig does not exhibit distinct seasonality in its breeding patterns, a noticeable trend exists toward reduced fertility in autumn. In Europe, seasonal infertility occurs in late summer and early autumn [15]. Nonetheless, the severity of seasonal infertility in domestic pigs further depends on external factors, which include the feeding level and group housing [17].

In domestic pigs, the maximum concentrations of steroids coincide with the breeding season of wild pigs in the same latitudes [18]. These fluctuations appear to be a key driver of seasonal changes in testicular function in domestic boars, especially those reared outdoors under natural lighting. Notably, androstenone concentrations in the fat, blood, and semen of boars are about five-fold higher from October to December, compared to the rest of the year [18], and these concentrations respond to changes in day length. The role of photoperiod and hormones in these seasonal variations is prominent, with testicular activity and related male characteristics reaching a maximum during times when day length is less than 12 h [19]. Seasonal fluctuations also extend to sperm production, with declines and poorer sperm quality observed in summer, taking several months to return to baseline levels [20].

This study aimed to compare the testicular morphometry and meat quality traits, including boar taint compounds, in Bísaro entire males slaughtered in different seasons (winter and summer).

## 2. Materials and Methods

The research took place at a Bísaro commercial pig farm located in northern Portugal (41°22′41.8″ N; 6°51′17.0″ W). All the animals were managed in accordance with the national [21] and European regulations governing animal welfare in experimental research [22].

### 2.1. Animal Husbandry

A total of 20 non-castrated male Bísaro pigs were used in this study. The animals were divided into two groups according to the season at slaughter: Su (*n* = 11), comprising those slaughtered in summer, and Wi (*n* = 9), encompassing pigs slaughtered in winter. The experimental design, along with the mean maximum and minimum ambient temperatures, are represented in Figure 1.

Both Su and Wi groups were housed indoors until 4 months old. In November, the Su group moved into outdoor pens with natural day-length conditions (see Figure 2) until slaughter at 13 months old (August). The Wi group was kept in outdoor pens from May until slaughter at 13 months old (February). All animals adhered to a diet of the same commercial feed and received ad libitum supplementation with sugar beets (*Beta vulgaris* L.) all year round, following the regular farm routine. The outdoor pens had *Quercus* spp., providing free acorn supplementations in the winter months. No deaths were recorded in this study.

The animals were transported to an official slaughterhouse according to the current legislation on welfare and animal-derived food product hygiene [23,24,25]. The animal carcasses were closely monitored along the slaughter line until evisceration. Besides a portion of subcutaneous fat from the neck region, the gonads were individually collected and placed in labeled containers to ensure traceability. The samples were kept at 4 °C and later transported to the laboratory for analysis. The carcasses were split longitudinally, and the corresponding hot carcass weights were documented for carcass yield calculation. The evaluation of meat quality used the Longissimus thoracis et lumborum (LTL) muscle, collected from the left side of the carcasses.

### 2.2. Reproductive Tract Morphometry

The gonads were subject to visual inspection at the Laboratory of Histology and Anatomical Pathology (LHAP) from the University of Trás-os-Montes and Alto Douro (UTAD), Vila Real (Portugal), and any abnormality documented. Their weight and measurements were obtained as per Paixão et al. [12]. The aggregated weight and length of the testis and epididymis were recorded before the testis and epididymis were carefully separated and measured individually. The testicular length, width, height, and weight were measured (Figure 3). The testicular volume was estimated using the empiric Lambert formula (testicular length × width × depth × 0.71) [26]. Epididymes were measured in length and width at the head, body, and caudal regions.

The gonadosomatic index (GSI) was calculated by applying the formula (testis weight/body weight × 100), providing insight into testicular development and reproductive potential [27].

### 2.3. Physicochemical Analysis

All procedures in this study followed the methodology described previously [28]. A pH meter WTW 330i (Weilheim, Germany) was used to record carcasses’ pH at 45 min postmortem (pH_45min_), while still in the slaughterhouse line. The instrument was previously calibrated with pH 4.01 and 7.00 buffers. Duplicate readings were obtained by inserting the probe between the 13th and 14th thoracic vertebrae. A sample from the LTL muscle (1.5 kg) was excised 24 h postmortem, between the 7th thoracic vertebrae and the 3rd lumbar vertebrae. The samples were then refrigerated at 4 °C and transported to the Laboratório de Tecnologia, Qualidade e Segurança Alimentar (TeQSA) at the University of Trás-os-Montes and Alto Douro (UTAD), Vila Real (Portugal). Further processing included fat and connective tissue trimming and slicing for subsequent analysis. Meat quality attributes evaluated in the fresh cuts comprised pH_24h_, color coordinates (*L**, luminosity; *a**, red–green; *b**, yellow–blue; C*, chroma; and h°, hue angle), drip loss, cooking loss, and shear force. Moreover, approximately 100 g and 400 g samples were vacuum-packed and stored at −20 °C for subsequent chemical analyses and sensory evaluation, respectively. The portion of subcutaneous neck fat collected was also stored at −20 °C specifically for the quantification of boar taint compounds, androstenone, and skatole.

The ultimate pH (pH_24h_) was measured in duplicate 24 h postmortem using the same equipment used for the evaluation of pH_45min_. Color coordinates *L**, *a**, *b**, C*, and h° [29] were assessed by measuring a 2 cm thick meat slice using a Minolta Chroma Meter CR-310 colorimeter (Osaka, Japan), after 60 min of blooming, at 4 °C. The colorimeter underwent calibration using a standard white ceramic plate with a D65 illuminant observer angle of 0°, and an aperture size of 5.0 mm. Heme pigments were extracted according to Hornsey [30] and multiplied by 0.026 [31] to express data in total heme pigments (mg/g). Drip loss was quantified as a percentage of mass loss relative to the initial mass of the sample using the suspension method of Honikel [32]. Cooking loss was evaluated by heating samples with similar shape and weight in a water bath at 80 °C until an internal temperature of 75 °C was reached and then cooled in an ice bath until 4 °C. The samples were then dried with filter paper and weighed. The cooking loss was calculated as the percentage of mass lost compared to the initial sample mass [33]. Afterward, the samples were stored overnight at 4 °C for subsequent shear force measurements. Cuboid subsamples, consisting of four to six slices, were carefully prepared, and after equilibrating to room temperature their shear force was evaluated using a Warner–Bratzler rectangular hole probe connected to a TA.XT.plus texturometer (Stable Micro Systems, Godalming, UK) with a 30 kg load cell, blade velocity 200 mm/min, and trigger force of 5 g. The recorded maximum shear force values were expressed in N/cm^2^.

The approximate chemical composition, expressed as a mass percentage, encompassed moisture, fat, protein, and ashes determination. The moisture content was assessed following ISO 1442:1997 [34]. The lipid content was determined according to the American Oil Chemists’ Society (AOCS) Official Procedure Am 5-04, employing the Ankom XT10 fat extractor (ANKOM Technology Corp., Macedon, NY, USA). The total nitrogen content was determined using the Kjeldahl method. For the total nitrogen content determination, the Kjeldahl method was used [35], and converted to protein using the factor 6.25. The ash content was determined according to ISO 936:1998 [36].

Androstenone (AND) and skatole (SKA) levels in pig fat were quantified using HPLC analysis following the procedures described earlier [28]. The limits of detection (LoD) of the assays were 16.02 ng/mL and 1.53, respectively, for androstenone and skatole, and the recovery values were 102.84% for androstenone, and 99.72% for skatole. Method validation included repeatability <2.46% RSD for SKA and <6.85% RSD for AND, as well as the intermediate precision, which was <2.87% RSD for SKA and <6.98% RSD for AND [37].

### 2.4. Sensory Analysis

The selection and training of the sensory panelists were described before [38]. Briefly, the sensory performance of twenty-two adults was assessed following the ISO 8586:2012 [39] methodology over five sessions; ten panelists were excluded because of their low performance. The remaining 12 panelists underwent training following a procedure adapted from Garrido et al. [40]. Standard solutions for AND (5α-androst-16-en-3-one, M 272.43 g mol^−1^, Sigma-Aldrich A8008, Saint Louis, MI, USA) and SKA (3-methylindole, M 131.17 g mol^−1^, Sigma-Aldrich M51458, Saint Louis, MI, USA) were prepared using Vaseline oil. Panelist training involved various tests, including descriptive, ranking, classification, and triangular tests, using concentrations as described in Garrido et al. [40] throughout 16 sessions. Additionally, training sessions incorporated loins from pigs with known levels (low and high) of boar taint compounds over five sessions.

Twenty-four hours before the sensory evaluation, the meat samples were thawed at 4 °C. Four animals from each group were randomly selected, and the loin fillets (1.5 cm) were prepared and served following the instructions outlined in Silva [41]. To maintain the original boar taint unaltered [40,42], no extra seasoning was added. Subsequently, two 2 × 2 cm pieces of meat were extracted from the cooked fillets, wrapped in aluminum foil individually, and maintained at 60 °C until evaluation. They were then assigned a distinct three-digit identifier, and the presentation order of the samples was randomized to minimize the risk of any potential sequencing bias. The sensory analysis was carried out in a controlled sensory analysis laboratory, equipped with individual booths, ensuring consistent illumination conditions and room temperature ranging between 18 and 25 °C. Palate cleansers, consisting of room-temperature spring water and bread, were provided between samples. Assessors were directed to assess the aroma and taste of the samples based on their overall flavor intensity (boar taint, urine, and manure) and texture (juiciness, tenderness, and chewiness) using a scale from 1 to 9, without intermediate values.

### 2.5. Fatty Acid Profile

The protocol for fat extraction and transesterification used was defined by Domínguez et al. [43] for fatty acid determination. For the separation and quantification of fatty acid methyl esters (FAMEs), a gas chromatograph (Agilent DB-23; Agilent Technologies, Santa Clara, CA, USA), equipped with a flame ionization detector (FID), and a PAL RTC-120 autosampler, with a liquid injection tool (Pal System), were used. The chromatographic conditions followed those recommended by Domínguez et al. [43]. For the separation of FAMEs, a DB-23 fused silica capillary column (60 m, 0.25 mm i.d., 0.25 µm film thickness; Agilent Technologies) was employed. The results were expressed as grams per 100 g of fat. Furthermore, n − 6/n − 3 and PUFA/SFA ratios were determined [44].

### 2.6. Statistical Analysis

Statistical analysis was performed using IBM SPSS Statistics software (version 29) from IBM Corp (Armonk, NY, USA). The variables underwent testing to evaluate their distribution and data normality through the Shapiro–Wilk test. The live and carcass weight, dressing %, aggregated testis–epididymis weight, testicular width, height, weight and volume, epididymis length, width of the head and tail, epididymis weight, pH_45min_, color coordinates *a**, *b**, C*, and h°, heme, cooking loss, shear force, moisture, protein, ashes, aroma (pork intensity and urine), flavor, juiciness, tenderness, and chewiness, seeing as they belong to the normal distribution, were analyzed using one-way ANOVA. Nonetheless, the aggregated testis–epididymis weight, testicular length, pH_24h_, *L**, drip loss, intramuscular fat, androstenone, skatole, and aroma (boar taint and feces) variables did not fit the normal distribution. In such cases, non-parametric tests were performed. Significance was set at *p* < 0.05. The values expressed in the results are means to improve understanding, except in the graphs for androstenone and skatole, where values are expressed as median, Q1, Q2, max, and min. Significant differences were considered when *p* < 0.05.

## 3. Results

### 3.1. Reproductive Tract Morphometry

The gonad morphometry is presented in Table 1.

When analyzing the aggregated weight and length of the gonadal segment, no statistically significant variances were observed between the two groups (*p* > 0.05). However, the width and volume of the testes were notably lower in Su than Wi groups (*p* < 0.05). In contrast, regarding the measurements of the epididymis width in the head and caudal regions, Wi males displayed a broader epididymis head but smaller caudal regions (*p* < 0.001).

### 3.2. Physicochemical Analysis

The results for the carcass traits and physicochemical parameters are presented in Table 2.

Entire males slaughtered in winter (Wi group) exhibited reduced live weight compared to those slaughtered in summer (Su), resulting in lower carcass weight (*p* < 0.05). There were no significant differences in dressing percentage (*p* > 0.05). The impact of the season at slaughter was minimal on most meat quality attributes, with the exceptions being pH_45min_, pH_24h_, moisture, and protein content. Both pH_45min_ postmortem and pH_24h_ exhibited higher values in Wi boars than Su males and higher protein content. In contrast, the Su group demonstrated higher moisture content.

The results for the boar taint compounds are presented in Figure 4.

Although there were no significant differences regarding the existence of boar taint compounds (AND and SKA) between the two groups (*p* > 0.05), it is noteworthy that low SKA levels were found in groups; in contrast, the Su group displayed higher maximum AND levels compared to Wi’s.

### 3.3. Sensory Analysis

Sensory analysis results are presented in Table 3.

When evaluating the intensity of pork aroma and flavor in meat samples, panelists did not detect any significant differences between the groups under study (*p* > 0.05). Likewise, there were no discernible variations in urine and feces’ aroma and flavor attributes (*p* > 0.05). However, notable distinctions arose when assessing the boar taint characteristics in both aroma and flavor, with animals in Su group demonstrating a more pronounced boar taint feature than those in Wi (*p* < 0.05). On the contrary, no disparities were noted between the two groups concerning juiciness, tenderness, and chewiness (*p* > 0.05).

### 3.4. Fatty Acid Profile

The impact of the season on the fatty acid profile of LTL muscle is shown in Table 4.

Regarding the fatty acid composition, there were no significant differences in total saturated fatty acids (*p* > 0.05). However, the palmitic acid (C16:0) content was notably higher in Wi compared to Su (*p* < 0.05). Similarly, both the monounsaturated fatty acids (MUFA) and polyunsaturated fatty acids (PUFA) exhibited higher levels in the Wi group (*p* < 0.05), primarily due to the elevated presence of palmitoleic acid (C16:1 n − 7), arachidonic acid (C20:4 n − 6), eicosapentaenoic acid (C20:5 n − 3), and docosahexaenoic acid (C22:6 n − 3) (*p* < 0.05). The total n−3 and n−6 fatty acids also displayed higher values for pigs slaughtered in the winter (Wi), resulting in a higher n − 6/n − 3 ratio for pigs slaughtered in summer (Su). In summary, there were significant distinctions in the fatty acid profile between the Su and Wi groups, with Wi animals exhibiting higher levels of most assessed fatty acids.

## 4. Discussion

Traditional outdoor rearing is susceptible to the varying weather conditions and day lengths associated with each season. The present study found no differences in testicular weight, although some studies performed on wild boars have found a marked seasonal pattern of plasma testosterone concentrations, paralleling the significantly increased testicular weight [19] and dimensions in winter, specifically the testicular weight and length, and the epididymal weight [20]. Seasonal influence on testicular steroids is common for wild animals [18]. Wild boars, for instance, exhibit short-day breeding behavior, where testicular activity increases when the day length is less than 12 h [19]. In contrast, reproduction in the domesticated boar typically remains consistent throughout the year [19]. However, some studies have noted a decrease in testicular steroids during the summer in domestic boars, which seems to be influenced by both daylight [18] and temperature [45]. While several studies have investigated the impact of the season on semen production and viability in domestic boars [46,47,48,49,50,51,52,53,54], limited research has been conducted on gonad morphometry [45,55]. Testosterone levels offer a means to assess how the season affects the state of gonad development, due to its connection with the hypothalamic–pituitary axis [54]. Since the primary endocrine function of the testis is testosterone production, testosterone levels can be indicative of higher or lower function of the gonads. The differences in testicular volume observed in the present study might be related to an increase in testosterone production from October to December [18,54], along with an increased efficiency of the spermatogenic process, and are coincident with the wild boar breeding season [18]. Despite the winter slaughter of Wi pigs, the actual slaughter took place in February. This timing might account for the lack of differences among groups in terms of most testicular dimensions, except for testicular width. It is possible that the animals were undergoing recovery from the impact of the photoperiod on their morphometry [51,54,56]. It has been shown that elevated testosterone levels are associated with an increase in testicular responsiveness and size [45]. Regarding the epididymal measurements, our study found contrasting results for the width of the epididymal head and tail, with Wi having a larger head width, but a smaller tail width, than Su. There were no differences in length or weight, although boars with higher testosterone levels are also associated with higher epididymal weights, while the decrease in testosterone leads to degeneration of the epididymal epithelium [55]. The region of the head is associated with spermatozoa maturation, while the caudal region is for storage of mature spermatozoa before ejaculation [57], which could explain why the head width was larger, due to higher testicular function, but it cannot explain the reason for the tail to be smaller.

The effect of season on carcass characteristics and meat quality traits has not been extensively studied. Most studies compare the indoor and outdoor production systems but fail to take into accordance the season when the study is being performed. However, the ambient temperature to which the pigs are exposed has already been extensively studied [58,59,60,61,62,63]. The ambient temperature is a factor that affects the metabolic pathways for thermoregulation and feed intake, and, in consequence, fat deposition and fatty acid composition [61]. Because a decrease in ambient temperature is often associated with an increase in voluntary feed intake [61], pigs slaughtered in winter usually have higher live weight and carcass weight than those slaughtered in summer [64]. This was not verified in our study. Our results presented 10.85% higher live and carcass weights for animals slaughtered in summer (Table 2). A possible explanation is the farm management, where the pigs are kept indoors until 4 months old, and then are sent to outdoor pens. The pigs that were slaughtered in the summer (Su) were kept in indoor pens until the middle of winter, while the pigs slaughtered in the winter (Wi) were kept indoors until the beginning of summer. This suggests that the Su group took advantage of the increase in voluntary feed intake, increasing their growth rate, while the Wi group was put in outdoor pens in summer when temperatures were compatible with thermal stress, which is known to have the lowest average growth rates [63]. Some authors also reported decreased growth rates in the winter in outdoor pastures, possibly due to higher nutrient requirements due to increased heat production to maintain body temperature [65,66].

The mean values of pH_45min_ and pH_24h_ are higher in Wi than Su (*p* < 0.05), which is in accordance with Čobanović et al. [64]. Low ambient temperature, below the thermoneutral zone, increases muscle glycogen stores leading to lower pH_24h_ [59]. The pH is a critical factor in assessing meat quality, particularly in the evaluation of PSE (pale, soft, and exudative) and DFD (dark, firm, and dry) meats [67,68]. Neither PSE nor DFD meats were found in the present study, due to the absence of pH_45min_ below 5.8 and pH_24h_ higher than 6.0, respectively [67,68]. High ambient temperatures can lead to acute stress, which in turn increases the rate of acidification of the muscles immediately after death [67,68]. This leads to low pH values in the early postmortem period, such as presented in the Su group. Lower pH_45min_ also leads to a reduced water-holding capacity, which leads to an increase in drip loss and *L** parameter [64,67,68], which is in accordance with our findings (despite no significant differences). Contradictory results have been found regarding these parameters, where some authors found that pigs slaughtered in winter have higher occurrences of DFD meat [69,70,71], and others found that there was a higher prevalence of RSE meats [64]. Neither of these cases was found in the present study. Regarding the color parameters, no marked differences were found, although the *L** was numerically higher in Su [64], which is in accordance with the observed pH values. In contrast, Lebret et al. [63] reported higher *L** and *b** values in winter than in summer. Water-holding capacity, which can be defined using drip loss and cooking loss, presented no significant differences between the two studied groups, as observed by Čobanović et al. [64]. On the other hand, Lebret et al. [63] found that drip loss was higher in winter, possibly due to higher lactate levels. The Wi group presented higher shear force values than the Su group (*p* > 0.05). These findings align with a positive correlation between shear force and intramuscular fat [72], though this relationship does not wholly account for the substantial numerical variations between groups, despite the absence of significant differences. Physical activity also exhibits a positive correlation with shear force [73], wherein animals engaged in higher physical activity, related to colder months, exhibit higher shear force values. In terms of chemical properties, significant differences were observed in both moisture and protein contents. The Su group presented higher moisture content than the Wi group, consistent with Barlocco et al. [72]. Protein content aligns with the remaining results, as meats with lower protein content usually have higher moisture content. While neither intramuscular fat nor ashes displayed significant differences, it is noteworthy that the Wi group presented numerically higher intramuscular values than the Su group, potentially due to a higher acorn content in their diet [5].

Although no significant differences were observed between the studied groups concerning the boar taint compounds, there appears to be a tendency toward higher values in the Su group compared to the Wi group. So far, findings related to the impact of season on androstenone levels have yielded contradictory results. Some authors found no seasonal influence on fat androstenone [74], whereas others noted higher androstenone levels during winter [75]. Walstra et al. [76] observed higher androstenone levels in winter in the Netherlands, while in the UK higher values were found during the summer. In contrast, skatole findings are more consistent, some authors reported higher levels in the summer [74,76,77]. Skatole can be reabsorbed through a pig’s skin upon contact with fecal contamination, thus increasing the fat skatole levels when the animal is in contact with fecal and urine residues [77]. Multiple studies have identified skatole concentration levels in intact males ranging from 100 ng/g to 200 ng/g of liquid fat [78,79,80,81,82,83,84,85,86]. In the present study, all values were lower than 80 ng/g, which might be attributed to the beet supplementation, a measure previously demonstrated to effectively reduce skatole levels [86,87]. The carbohydrates provided to the animals throughout the year in the form of beet pulp have the capacity to alter the activity and composition of gut microflora. This modification results in a diminished availability of L-tryptophan, the precursor to skatole, in the colon, driving reduced skatole levels in pig fat [86]. This reduction may be initiated through one of three potential pathways: fermentation of pulp generating butyrate resulting in a decrease in local tryptophan availability; an overall boost in microbial activity causing tryptophan incorporation into the microbial mass; alteration in the microbiota composition, prompting a reduction in skatole-producing bacteria within the intestinal lumen [88,89].

To our knowledge, there are no studies performed on the effect of season on the sensory quality of pork. The panelists found significant differences between the Su and Wi groups regarding the boar taint (aroma and flavor), confirming the data from the boar taint compounds analysis: higher intensity of boar taint in the Su group. Although no significant differences were found in juiciness, tenderness, and chewiness, the panelists’ findings are in accordance with the meat quality parameters studied.

The fatty acid composition in pork is heavily influenced by the pig’s diet, but it could also be influenced by their genetics [61,90]. While saturated fatty acids (SFA) are primarily derived from de novo synthesis, polyunsaturated fatty acids (PUFA) cannot be synthesized by the pig’s digestive system, instead originating from their diet [91]. Monounsaturated fatty acids (MUFA) can be acquired through both means [91]. Therefore, it is highly recommended that animal feed has ingredients with a high PUFA profile, mainly due to their benefits on animal health and meat quality [92]. The diet of the studied pigs was mainly a commercial diet, but it was also supplemented with beets (without fatty acid content) and acorns (seasonal availability, only in winter months). Acorns are a rich source of fat and have other compounds, such as α-tocopherol, γ-tocopherol, or tannins, which contribute to preventing lipid peroxidation [13]. Several studies have been performed on the effect of the addition of acorns to the pig’s diet [2,5,6,93,94]. The present study found no significant differences in the total SFA content, although there is a tendency for higher content in the Wi group (*p* = 0.063). These higher values can be attributed to the higher content of palmitic acid (C16:0) (*p* < 0.05). Although palmitic (C16:0) and stearic (C18:0) acids are the acids that predominate the SFA group [95], our results are not in accordance with the studies performed so far [2,5,6,93,94], which have found lower SFA content in pigs fed with acorns. Stearic acid (C18:0) also has a large influence on meat firmness [96], which is in accordance with the high values of shear force for the Wi group, despite not having significant differences between groups. Total MUFA presented significantly higher values for the Wi group than the Su group (*p* < 0.05), which is in accordance with several studies [2,5,6,93,94]. Although those studies accrue the increase in MUFA to the increase in oleic acid (C18:1 n − 9), the present study did not find significant differences in that fatty acid (*p* = 0.074). Differences were found in palmitoleic acid (C16:1 n − 7), which is derived from palmitic acid (C16:0), with the Wi group having higher values than the Su group. Total PUFA content was significantly higher in the Wi group, due to the higher levels of arachidonic acid (C20:4 n − 6) (*p* < 0.05) and linoleic acid (C18:2 n − 6) (*p* = 0.082). Results are controversial, as some authors report higher PUFA content [2] while other studies found lower total PUFAs [5,94], which might be due to the different diets or times of the year when the studies were conducted. Higher levels of eicosapentaenoic (C20:5 n − 3), docosapentaenoic (C22:5 n − 3), and docosahexaenoic (C22:6 n − 3) acids were found in the Wi group (*p* < 0.05), which belong to the PUFA n − 3 content that several authors reported high values from [2,6,94], and which this study also presents. Experts recommend a diet rich in PUFAs and MUFAs, while limiting SFAs due to their link to cardiovascular diseases [97,98]. The health benefits of a product are largely determined by its fatty acid profile. One way to measure this is by using the atherogenic index (AI), which indicates the proportion of saturated to unsaturated fatty acids in the product. Lower values of AI indicate a lower content of saturated acids, which could lower the risk of atheroma development in humans [99]. Although no significant differences were observed, the values are low, making the meat from intact Bísaro pigs a healthy product in this sense.

## 5. Conclusions

The seasonality influenced the testicular morphology in such parameters as volume, giving the indication of a diminished function in summer. There were no marked differences in meat quality traits, and although there were no statistical differences in boar taint compounds, in the sensory evaluation the panelists were able to detect a stronger boar taint aroma and flavor in the animals from the Su group. Fatty acid content (MUFA and PUFA) was higher in the Wi group, which could be attributed to the differences in their diet (acorns in winter months).

## Figures and Tables

**Figure 1 animals-14-00102-f001:**
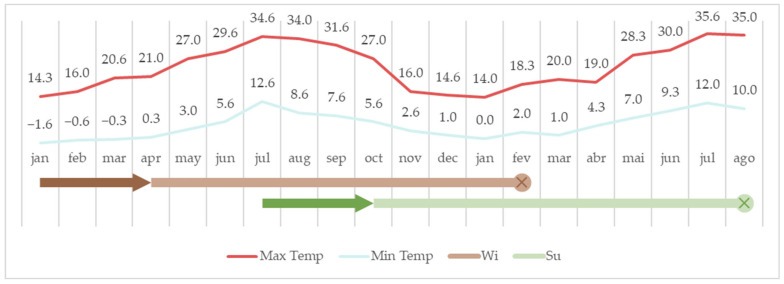
Mean maximum and minimum monthly ambient temperatures throughout the study. The arrow represents the period of indoor housing, before the transition to outdoor pens, and the round mark with an X signals the slaughter.

**Figure 2 animals-14-00102-f002:**
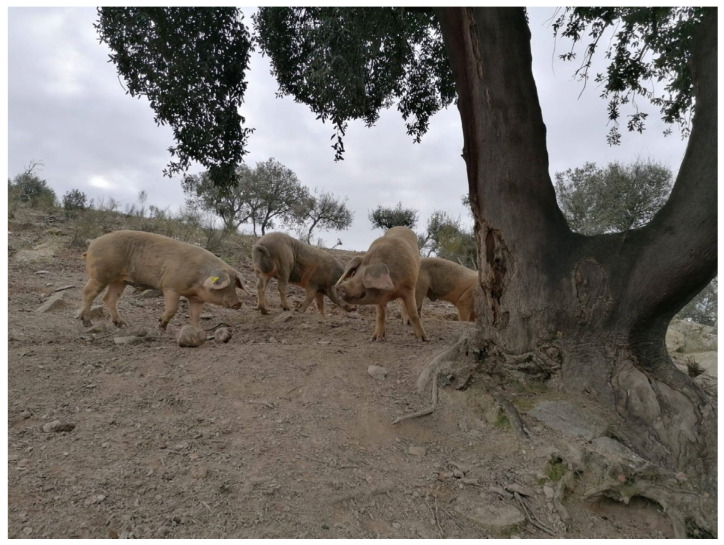
Traditional outdoor pen in Bísaro pig production.

**Figure 3 animals-14-00102-f003:**
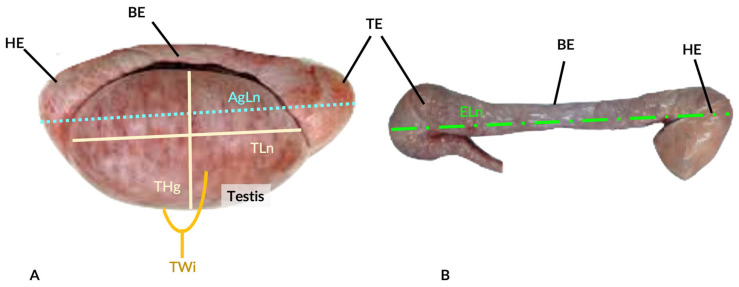
Graphic representation of the anatomic limits used for the morphometric measurements in this study: (**A**) The aggregated length (AgLn) of the testis and epididymis; testicular length (TLn), testicular height (THg), and testicular width (TWi). (**B**) The epididymis length (ELn). HE—head of epididymis; BE—body of epididymis; TE—tail of epididymis.

**Figure 4 animals-14-00102-f004:**
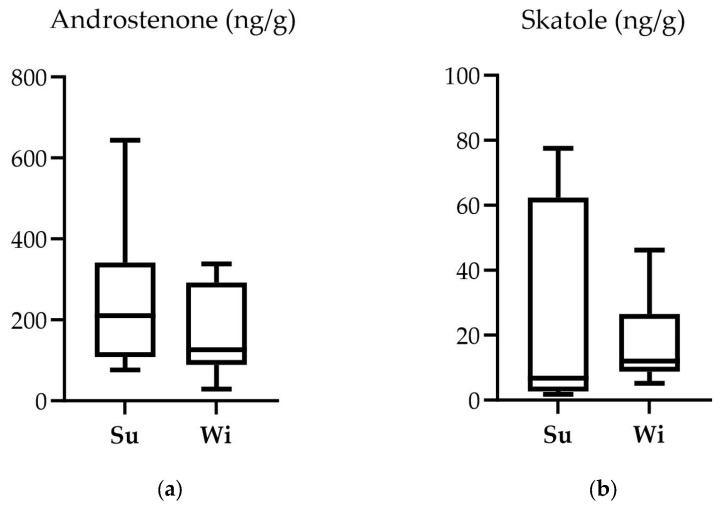
Boar taint compounds content (ng/g) in the neck fat of boars. (**a**) Androstenone, Su 210.38 ng/g (median), Wi 126.42 ng/g (median), SEM 33.74, *p* = 0.261; (**b**) Skatole, Su 6.76 ng/g (median), Wi 12.02 ng/g (median), SEM 5.19, *p* = 0.456.

**Table 1 animals-14-00102-t001:** Gonad morphometry.

Parameter	Su (*n* = 11)	Wi (*n* = 9)	SEM	*p*-Value
Aggregate				
Weight (g)	829.27	834.00	27.47	0.824
Length (cm)	16.77	17.50	0.25	0.145
Testicles				
Length (cm)	12.36	12.86	0.21	0.331
Width (cm)	7.68	8.81	0.21	**0.005**
Height (cm)	6.50	6.67	0.11	0.462
Weight (g)	313.05	312.06	10.95	0.966
Volume (cm^3^)	444.07	538.90	22.35	**0.031**
Epididymis				
Length (cm)	20.11	21.19	0.30	0.072
Width of the head (cm)	4.14	5.06	0.15	**<0.001**
Width of the tail (cm)	4.66	3.42	0.19	**<0.001**
Weight (g)	101.59	104.94	3.66	0.661
GSI	0.48	0.54	0.02	0.152

Su: Boars slaughtered in Summer; Wi: Boars slaughtered in Winter. All values are means. SEM: standard error of the mean.

**Table 2 animals-14-00102-t002:** Carcass traits and physicochemical parameters.

Parameter	Su (*n* = 11)	Wi (*n* = 9)	SEM	*p*-Value
Live weight (kg)	171.35	153.71	3.57	**0.010**
Carcass weight (kg)	136.04	118.17	3.45	**0.006**
Dressing (%)	79.33	76.85	0.68	0.067
pH_45min_	6.15	6.41	0.06	**0.029**
pH_24h_	5.41	5.60	0.03	**<0.001**
*L**	51.41	47.86	0.95	0.800
*a**	21.94	21.91	0.26	0.968
*b**	6.25	6.43	0.27	0.757
C*	22.83	22.86	0.30	0.955
h°	15.89	16.22	0.55	0.772
Heme (mg/g)	1.60	1.71	0.05	0.241
Drip loss (%)	4.15	3.05	0.33	0.131
Cooking loss (%)	27.75	28.02	0.63	0.840
Shear force (N/cm^2^)	54.95	71.20	4.21	0.052
Moisture (%)	73.08	72.12	0.18	**0.004**
Protein (%)	22.79	23.44	0.12	**0.004**
Intramuscular Fat (%)	1.50	1.93	0.18	0.261
Ashes (%)	1.14	1.13	0.01	0.635

Su: Boars slaughtered in Summer; Wi: Boars slaughtered in Winter. All values are means. SEM: standard error of the mean.

**Table 3 animals-14-00102-t003:** Sensory analysis of the LTL muscle in entire males.

Parameter	Su (*n* = 4)	Wi (*n* = 4)	SEM	*p*-Value
Aroma				
Pork Intensity	5.4	5.4	0.2	0.865
Boar taint	4.9	3.3	0.3	**0.029**
Urine	3.4	2.6	0.3	0.091
Feces	2.8	2.2	0.3	0.343
Flavor				
Pork Intensity	5.2	5.4	0.1	0.366
Boar taint	3.7	2.7	0.2	**<0.001**
Urine	2.2	2.2	0.2	0.916
Feces	1.5	1.5	0.1	0.958
Juiciness	4.8	5.2	0.2	0.449
Tenderness	5.2	5.4	0.3	0.771
Chewiness	5.1	5.4	0.2	0.548

Su: Boars slaughtered in Summer; Wi: Boars slaughtered in Winter. All values are means. SEM: standard error of the mean.

**Table 4 animals-14-00102-t004:** Fatty acid profile (g/100 g of pork) of LTL muscle.

Fatty Acid (mg/100 g Pork)	Su (*n* = 11)	Wi (*n* = 9)	SEM	*p*-Value
Saturated fatty acid (SFA)	708.03	903.71	72.23	0.063
Lauric acid (C12:0)	1.29	1.42	0.14	0.909
Myristic acid (C14:0)	23.27	27.49	2.48	0.790
Pentadecylic acid (C15:0)	0.55	0.50	0.05	0.849
Palmitic acid (C16:0)	439.69	582.43	46.08	**0.037**
Heptadecanoic acid (C17:0)	3.50	3.74	0.30	0.732
Stearic acid (C18:0)	233.02	279.79	22.76	0.239
Arachidic acid (C20:0)	3.56	4.74	0.40	0.102
Monounsaturated fatty acid (MUFA)	908.24	1232.13	102.31	**0.037**
Palmitoleic acid (C16:1 n − 7)	59.70	89.01	8.08	**0.020**
Oleic acid (C18:1 n − 9)	733.02	977.13	83.39	0.074
Eicosenoic acid (C20:1 n − 9)	12.70	15.42	1.18	0.305
Polyunsaturated fatty acid (PUFA)	291.85	354.23	14.00	**0.022**
Linoleic acid (C18:2 n − 6)	220.29	259.35	11.19	0.082
Alpha-linolenic (C18:3 n − 3)	5.20	6.15	0.44	0.210
Dihomo-γ-linolenic acid (C20:3 n − 6)	5.87	5.90	0.20	0.949
Arachidonic acid (C20:4 n − 6)	47.09	64.13	2.73	**<** **0.001**
Eicosatrienoic acid (C20:3 n − 3)	1.14	1.27	0.07	0.342
Eicosapentaenoic acid (C20:5 n − 3)	0.78	1.55	0.10	**<** **0.001**
Docosapentaenoic acid (C22:5 n − 3)	3.39	6.32	0.42	**<** **0.001**
Docosahexaenoic acid (C22:6 n − 3)	0.15	0.84	0.12	**0.004**
PUFA/SFA	0.44	0.42	0.02	0.729
Total n − 3	10.66	16.14	0.87	**<** **0.001**
Total n − 6	281.17	338.09	13.24	**0.028**
n − 6/n − 3	27.26	21.03	1.09	**<** **0.001**
Index of atherogenicity	0.44	0.43	0.01	0.458

Su: Boars slaughtered in Summer; Wi: Boars slaughtered in Winter. All values are means. SEM: standard error of the mean.

## Data Availability

The data presented in this study are available on request from the corresponding author. The data are not publicly available due to [commercial reasons].

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
