# Peer review of "Seasonal Variations on Testicular Morphology, Boar Taint, and Meat Quality Traits in Traditional Outdoor Pig Farming"

_animals, 2023, doi:10.3390/ani14010102_

Round 1

Reviewer 1 Report

Comments and Suggestions for Authors

It is not clear that how experimental groups were divided. If they were slaughtered in the same age, than they had to be housed outdoor in different time, which results in different feeding pattern (acorn access mainly).

Intramuscular fat content is surprisingly low. Modern genotypes having 1.2-1.3%, Duroc 4-5 %, while Mangalica (a comparable native breed) 7-8% usually. Why this difference? Discuss it please.

 Are the differences in fatty acid profile is due to the acorn intake? Discuss it.

line 19: when this breed reaches the breeding age? Can they be considered as mature animals?

Are they usually raised without castration for meat production?

line 22: please specify here that what type of tissue was analyzed for boar taint.

line 25: shedding light seems to be a second factor, but not mentioned earlier (nor later).

line 28: I would suggest to use the term „special quality”, as modern genotypes produce also high quality high quality meat.

line 98-99: Were all the animals housed outdoor at the same time? Provide month.

line 100: it reads, that the animals were at different age when slaughtered. Give exact month of slaughter, and explain more in detail the animal setup.

line 103: what beets? give scientific name, and proximal quantity offered by animal per day.

line 112: what fat was collected? Subcutaneous? What was the thickness of fat in that region? Why this place was selected? For what these samples were used (no results presented)?

line 113: better say:  individually marked (or labelled) containers

line 116: what amount of muscle was collected?

Table 1 and 2: it would be more logical to me to present carcass traits first and then gonad morphometry second.

Table 4: the unit in the title says g/100g of pork, then this is a kind of %. How saturated fat can be numerically 700-900, or MUFA over 1000? Please correct the unit and/or values.

line 329-330: the term winter is to wide here, specify month.

line 343: season effect can be considered independently if feeding is the same.

line 350: if exact value is given, then it is not about

line 351: data shown in table 2.

lines 351-355: Now it seems clear that the experimental animals have been put on outdoor field in different month. This has to be clearly described in materials and methods.

line 359: give information about winter temperatures where this experiment was run.

line 374: I guess PSE instead of RSE.

line 409: skatole level related to TRP metabolism. Discuss this in relation to feeding of these animals.

conclusions: In my opinion it is not correct to say that fatty acid content was higher in winter. It is not true for all FA, be more specific.

Author Response

We sincerely appreciate the valuable time and effort you dedicated to reviewing this manuscript. Below, you will find detailed responses to your feedback, along with the corresponding revisions and corrections, which have been highlighted or tracked in the resubmitted files.

Suggestions and comments:

It is not clear that how experimental groups were divided. If they were slaughtered in the same age, than they had to be housed outdoor in different time, which results in different feeding pattern (acorn access mainly). They were housed in different time periods thus having different feeding patterns, but slaughtered at the same age. This was rewritten in the manuscript in order to improve understanding: The animals were divided into two groups according to the season at slaughter: Su (n=11), comprising those slaughtered in summer, and Wi (n=9), encompassing pigs slaughtered in winter. The Su group was housed indoors until 4 months old (November) and kept in outdoor pens with natural day-length conditions (see Figure 1) until slaughter at 13 months old (August). The Wi group was also kept indoors until 4 months old (May) and kept in outdoor pens until slaughter at 13 months old (February).

Intramuscular fat content is surprisingly low. Modern genotypes having 1.2-1.3%, Duroc 4-5 %, while Mangalica (a comparable native breed) 7-8% usually. Why this difference? Discuss it please. Although some studies did find higher IMF content in Bísaro pigs than our results, some authors did present results similar to ours:

Álvarez-Rodríguez, J., & Teixeira, A. (2019). Slaughter weight rather than sex affects carcass cuts and tissue composition of Bisaro pigs. Meat science154, 54-60.

Teixeira, A., Silva, S. R., Hasse, M., Almeida, J. M., & Dias, L. (2021). Intramuscular fat prediction using color and image analysis of Bísaro pork breed. Foods10(1), 143.

The differences can be mainly due to the fact that most pigs are surgically castrated, which is proven to increase the IMF content significantly, and the housing systems. Pigs housed indoors also have higher IMF content than those raised outdoors.

 Are the differences in fatty acid profile is due to the acorn intake? Discuss it. The differences in fatty acids are mainly due to differences in the diet. Although the feed intake was not assessed due to the housing of the animals, it is known that they were fed ad libitum with a commercial diet, with the same composition for all animals, and sugar beets. The acorns were a feed that was only present in the winter months, so the conclusion reached is that the differences in fatty acid profile in the present study might be mainly due to the acorn intake.

line 19: when this breed reaches the breeding age? Can they be considered as mature animals?Are they usually raised without castration for meat production? This is a very precocious breed, reaching puberty between 3,5 to 4,5 months old. From that moment on, they are considered mature. Usually, male Bísaro pigs are surgically castrated in the first week, but some smaller farms miss that time frame, making it difficult to surgically castrate the animals. Those are usually used for meat products, in order to mask the boar taint.

line 22: please specify here that what type of tissue was analyzed for boar taint. This is explained in the material and methods.

line 25: shedding light seems to be a second factor, but not mentioned earlier (nor later). Rewritten in the manuscript for better understanding.

line 28: I would suggest to use the term „special quality”, as modern genotypes produce also high quality high quality meat. Rewritten in the manuscript for better understanding.

line 98-99: Were all the animals housed outdoor at the same time? Provide month. Rewritten in the manuscript for better understanding.

line 100: it reads, that the animals were at different age when slaughtered. Give exact month of slaughter, and explain more in detail the animal setup. Rewritten in the manuscript for better understanding.

line 103: what beets? give scientific name, and proximal quantity offered by animal per day. Information added in the manuscript.

line 112: what fat was collected? Subcutaneous? What was the thickness of fat in that region? Why this place was selected? For what these samples were used (no results presented)? The subcutaneous fat from the neck region was collected for boar taint compounds quantification. The fat thickness was not assessed. This place was selected because it was recommended in the protocol.

line 113: better say:  individually marked (or labelled) containers. Suggestion accepted, rewritten in the manuscript for better understanding.

line 116: what amount of muscle was collected? Information added in the manuscript for better understanding in line 146.

Table 1 and 2: it would be more logical to me to present carcass traits first and then gonad morphometry second. If the reviewer deems it necessary, the order of the paper may be altered. However, we suggest that changing the title of the paper may also be necessary to maintain coherence between the title and the results presented.

Table 4: the unit in the title says g/100g of pork, then this is a kind of %. How saturated fat can be numerically 700-900, or MUFA over 1000? Please correct the unit and/or values. The units were wrong, it was mg/100g. Thank you so much for pointing it out.

line 329-330: the term winter is to wide here, specify month. Suggestion accepted, rewritten in the manuscript.

line 343: season effect can be considered independently if feeding is the same. In fact, the entirety of the effects cannot be associated with the photoperiod, because the season also changes the animals' energy requirements and the diet itself (availability of certain foods). In this study, the feed intake was not assessed due to the animal housing.

line 350: if exact value is given, then it is not about. Suggestion accepted, rewritten in the manuscript.

line 351: data shown in table 2. Suggestion accepted, rewritten in the manuscript.

lines 351-355: Now it seems clear that the experimental animals have been put on outdoor field in different month. This has to be clearly described in materials and methods. Suggestion accepted, rewritten in the manuscript.

line 359: give information about winter temperatures where this experiment was run. A figure in the material and methods was added to improve the understanding of the experimental conditions.

line 374: I guess PSE instead of RSE. The work cited refers to RSE.

line 409: skatole level related to TRP metabolism. Discuss this in relation to feeding of these animals. Rewritten in the manuscript for further understanding.

conclusions: In my opinion it is not correct to say that fatty acid content was higher in winter. It is not true for all FA, be more specific. Suggestion accepted, rewritten in the manuscript.

Reviewer 2 Report

Comments and Suggestions for Authors

In recent decades, consumers have demanded increased animal welfare while producing food animals.  However, intact male pigs have a more pungent smell because of the puberal changes in their reproductive function, which can be eliminated by castration. The study aimed to evaluate the effect of the seasons of outdoor fattened entire male Bisaro pigs on their reproductive organs, carcass traits, and meat quality, especially boar taints (androsterone and skatole) by sensory analysis.

The title of the manuscript and the content of the abstracts are clear. The scientific background of the topic is sufficiently detailed about the alternative fattening, indigenous breeds, seasonal variation of the reproductive activity, and boar taint.   

All the investigation protocols are fully described in Materials and Methods.  Only one question should be addressed to the Authors about the exact time of the slaughtering. Suppose the reproductive function of the animals is investigated. In that case, part of the summer and the winter seasons belong to the transition period between reproductively more active and slightly less active intervals. Furthermore, breed-specific differences could be detected in these patterns in pigs, which are relatively high between modern breeds, wild boars, and indigenous pig breeds. Maybe this context should also be considered to improve later in the discussion part. Besides these parameters, temperature affects all the investigated parameters in outdoor keeping, so minimum monthly average temperature changes should be enclosed.

Results are summarized in four tables and one figure. The description of them is entirely appropriate except after the fourth table. The results were oppositely explained than they are in the table (Su and Wi).

Discussions are coherent and compared to previous research findings. The conclusion is well-underlined and moderate.  

After minor corrections, it is recommended to publish.

Author Response

We would like to express our sincere appreciation to the reviewer for dedicating his/her time and expertise to thoroughly assess our manuscript. Your invaluable insights and constructive comments have greatly contributed to the enhancement of our work.

The exact time of slaughter was added in the manuscript, as well as a figure with the temperatures throughout the study (mean minimum and maximum monthly temperatures). Thank you so much for your input, it has greatly improved our manuscript.

In the fourth table the results were oppositely explained, I’m very sorry for that mistake. It has already been corrected.

Once again, thank you so much for the time you put into our manuscript.